# Advances in the Use of Deep Learning for the Analysis of Magnetic Resonance Image in Neuro-Oncology

**DOI:** 10.3390/cancers16020300

**Published:** 2024-01-10

**Authors:** Carla Pitarch, Gulnur Ungan, Margarida Julià-Sapé, Alfredo Vellido

**Affiliations:** 1Department of Computer Science, Universitat Politècnica de Catalunya (UPC BarcelonaTech) and Intelligent Data Science and Artificial Intelligence (IDEAI-UPC) Research Center, 08034 Barcelona, Spain; alfredo.vellido@upc.edu; 2Eurecat, Digital Health Unit, Technology Centre of Catalonia, 08005 Barcelona, Spain; 3Departament de Bioquímica i Biologia Molecular and Institut de Biotecnologia i Biomedicina (IBB), Universitat Autònoma de Barcelona (UAB), 08193 Barcelona, Spain; gulnur.ungan@autonoma.cat (G.U.); margarita.julia@uab.cat (M.J.-S.); 4Centro de Investigación Biomédica en Red (CIBER), 28029 Madrid, Spain

**Keywords:** machine learning, neuro-oncology, radiology, deep learning, data analysis pipeline, ultra-low field magnetic resonance imaging

## Abstract

**Simple Summary:**

Within the rapidly evolving landscape of Machine Learning in the medical field, this paper focuses on the forefront advancements in neuro-oncological radiology. More specifically, it aims to provide the reader with an in-depth exploration of the latest advancements in employing Deep Learning methodologies for the classification of brain tumor radiological images. This review meticulously scrutinizes papers published from 2018 to 2023, unveiling ongoing topics of research while underscoring the main remaining challenges and potential avenues for future research identified by those studies. Beyond the review itself, the paper also underscores the importance of placing the image data modelling provided by Deep Learning techniques within the framework of analytical pipeline research. This means that data quality control and pre-processing should be correctly coupled with modelling itself, in a way that emphasizes the importance of responsible data utilization, as well as the critical need for transparency in data disclosure to ensure trustworthiness and reproducibility of findings.

**Abstract:**

Machine Learning is entering a phase of maturity, but its medical applications still lag behind in terms of practical use. The field of oncological radiology (and neuro-oncology in particular) is at the forefront of these developments, now boosted by the success of Deep-Learning methods for the analysis of medical images. This paper reviews in detail some of the most recent advances in the use of Deep Learning in this field, from the broader topic of the development of Machine-Learning-based analytical pipelines to specific instantiations of the use of Deep Learning in neuro-oncology; the latter including its use in the groundbreaking field of ultra-low field magnetic resonance imaging.

## 1. Introduction

Although Machine Learning (ML) is entering a phase of maturity, its applications in the medical domain at the point of care are still few and tentative at best. This paradoxical contradiction has been explained according to several different factors. One of them is the lack of experimental reproducibility, a requirement in which ML models in health have been reported to fare badly in comparison to other application areas [1]. One main reason to explain this is the mismatch between a data-centered (and often data-hungry) approach and the scarcity of publicly available and properly curated medical databases, combined with a nascent but insufficient data culture at the clinical level [2]. Another factor has to do with regulatory issues of ML (and Artificial Intelligence in general) in terms of both lack of maturity and geographical heterogeneity [3]. Further elements hampering ML-based tools adoption include data leakage, dataset shift, required model recalibrations, analytical pipeline maintenance failures, or changing medical practice patterns, to name a few [4].

The field of oncological radiology (and neuro-oncology in particular) is arguably at the forefront of the practical use of ML in medicine [5], now boosted by the success of Deep-Learning (DL) methods for the analysis of medical images [6,7]. Unfortunately, though, imaging does not escape the challenges and limitations summarized in the previous paragraph. Central to them, what has been called the “long-tail effect” [8]: pathologies for which only small and scattered datasets exist due to the scarcity of clinical data management strategies (technically complex and expensive) at levels beyond the local (regional, national, international). Associated with this, we must account for the difficulty of achieving standardized labeling (annotation) of imaging databases. An example of how to deal effectively with these problems is Federated Learning, which was used in [9] to gather data from 71 sites from 6 continents, analyzed using ML to address a problem of tumor boundary detection for glioblastoma brain tumors. Please note that the resulting database includes 6314 cases, which is impressive for this medical domain but still modest from an ML perspective. The success of ML in oncological radiology, as summarily stated in [10], will depend on its ability to create value in the delivery of medical care in terms of “increased diagnostic certainty, decreased time on task for radiologists, faster availability of results, and reduced costs of care with better outcomes for patients”.

This paper surveys some of the most recent advances in the use of ML for the analysis of magnetic resonance imaging (MRI) data in neuro-oncology without trying to make an all-encompassing review out of it. Instead, we focus on the most rapidly developing area, which involves the use of methods from the DL family. The variety of approaches sprouting from this family of methods has shaken the standards of data pre-processing or feature engineering before modeling as such. For this reason, we proceed to address the review hierarchically, starting in Section 3 with the broader topic of the development of ML-based analytical pipelines, which addresses the data analysis process beyond specific models and in which we will provide examples from two promising feature engineering approaches, namely source extraction in the form of independent component analysis (ICA) and nonnegative matrix factorization (NMF), and radiomics. The review of DL methods for image data analysis as such is delivered in Section 4. As an addition to this section, we will discuss the potential uses of DL in the groundbreaking field of ultra-low field (ULF) MRI [11]. Before all this, the following section will provide some contextual basic definitions of neuro-oncology concepts and a description of the main challenges and open issues concerning the use of ML in this domain.

## 2. Open Problems in AI Applied to MRI Analysis

The open problems for the use of ML-based analytical processes in the field of MRI in neuro-oncology can be seen from different perspectives. The first one is the analytical problem itself, according to which the main division is into categorization and segmentation problems. The latter is commented on later in this section.

Categorization can, in turn, be split into diagnosis and prognosis. In diagnosis, the correlation between neuroimaging classifications and histopathological diagnoses was assessed in [12] based on the 2000 version of the WHO classification of brain tumors and in [13] based on the 2007 version. In both studies, the main finding was that the sensitivity was variable among classes, whereas specificity was in the range of 0.85–1. The most difficult categories to diagnose were the glioma subtypes. The study based on the 2000 classification [12] reported a sensitivity of 0.14 for low-grade astrocytoma and 0.15 for low-grade oligodendroglioma. In the study based on the 2007 classification [13], increased sensitivity for low-grade astrocytoma (0.56) was found, but sensitivity was still low for other low-grade gliomas (LGG) such as oligodendroglioma (0.26), or for anaplastic gliomas (astrocytoma, 0.17 or ependymoma, 0.00), and other classes in the long-tail such as meningiomas of grade II and III in aggregate (0.17), or subependymomas and choroid plexus papilloma (0.33 for both). The recently released 2021 WHO classification [14], which incorporates the genetic alterations, opens the door to the reevaluation of these baseline results to accurately estimate the added value of any clinical decision support system (CDSS) based on ML or radiogenomics, over the limits of radiological interpretation of imaging findings. It is reasonable to foresee that the problematic tumor categories will remain so, or even more challenging, given the enhanced stratification of the glial category (e.g., different mutations of IDH1/2, ATRX, TP53, BRAF, H3F3A, CDKN2A/B, TERT and MGMT promoters, EGFR amplification, GFAP, 1p/19q codeletion, etc.).

On the other hand, regarding follow-up, there is no standard of care for recurrent high-grade gliomas, and the currently accepted criteria to assess response are those established by the Response Assessment in Neuro-Oncology Working Group (RANO) [15], which deals with the pseudoprogression phenomenon, defined as the appearance of contrast-enhancing lesions during the first 12 weeks after the end of the concomitant treatment or when the lesion developed within the first 3–6 months after radiation therapy, if it is in the radiation field (inside the 80% isodose line), and especially if it presents as a pattern of enhancement related to radiation-induced necrosis enhancement [16]. Also, with pseudoresponse in those patients treated with antiangiogenics in countries where these are approved [17,18]. Antiangiogenic agents, like bevacizumab, are designed to block the VEGF effect. The mechanism of action may be related to decreased blood supply to the tumor and normalization of tumor vessels, which display increased permeability. These agents are associated with high radiologic responses if we evaluate only the contrast enhancement. The recently published RANO 2.0 criteria [19] refine the former RANO, distinguishing between high-grade and low-grade gliomas. RANO 2.0 also takes into account the IDH status to decide whether the surrounding non-enhancing region should be taken into account or not. In this sense, ML-based pipelines should ideally be designed to allow the evaluation of their added value with respect to medical guidelines for clinical decision-making.

Another viewpoint to approach the open problems in the field has to do with the fact that ML-based analysis is strongly dependent on data pre-processing and the post-processing of results.

A fundamental prerequisite for the successful application of DL models in brain tumor classification is the pre-processing of the MRI data. The key pre-processing steps for the harmonization of MRI data are as follows.
**Resampling**: MRI scans can exhibit variations in resolution and voxel sizes depending on the acquisition system. Resampling standardizes the resolution across the MRI images to ensure uniform dimensions.**Co-registration**: entails the alignment of MRI scans with a standardized anatomical template with the purpose of situating different scans within the same anatomical coordinate system.**Skull-stripping**: The main objective of the skull-stripping step is to efficiently isolate the cerebral region of interest from non-cerebral tissues, which enables DL models to focus exclusively on those brain tissues.**Bias Field Correction**: aims to rectify intensity inhomogeneities that are pervasive in MRI scans to guarantee uniformity in intensity values. The technique of choice for bias field correction is N4ITK (N4 Bias Field Correction) [20], which is an improved variant of the N3 (non-parametric nonuniform normalization) retrospective bias correction algorithm [21].**Normalization**: a technique adopted to rescale intensity values of MRI scans to a numeric range, rendering them consistent across the dataset. This process mitigates scale-related disparities. Two prominent approaches commonly applied to MRI data as input for DL models are min-max normalization and z-score normalization. Min-max achieves its goal by rescaling intensity values within MRI scans, spanning their range between 0 and 1. In contrast, z-score, often referred to as standardization, transforms the distribution of intensity values by centering it around a zero mean and standard deviation of value 1.**Tumor identification**: A critical and optional pre-processing step before the classification task involves identifying the tumor region of interest (ROI) through segmentation or by defining a bounding box that encompasses the tumor. Popular DL architectures, such as UNet [22], Faster-RCNN [23], and Mask-RCNN [24] are often employed to perform such segmentation or detection tasks.

The post-processing of results must often address the fact that the DL family of methods is, by their nature, an extreme case of black-box approach, a characteristic that may strongly hamper their medical applicability [25]. This limitation can be addressed using explainability and interpretability strategies; for further details on these, the reader is referred to [25].

## 3. Ml-Based Analytical Pipelines and Their Use in Neuro-Oncology

Ultimately, the whole point of using ML methods for data-based problems in the area of neuro-oncology is to provide radiologists with evidence-based medical tools at the point of care that can assist them in decision-making processes, especially with ambiguous or borderline cases. This is why it makes sense to embed these methods in Clinical Decision Support Systems (CDSS). A thorough and systematic review of intelligent systems-based CDSS for brain tumor analysis based on magnetic resonance data (spectra or images) is presented in this same Special Issue of Cancers [26]. It reports their increasing use over the last decade, addressing problems that include general ones such as tumor detection, type classification, and grading, but also more specific ones such as physicians’ alerting of treatment change plans.

At the core of ML-based CDSS, we need not just ML methods, models, and techniques but, more formally, ML pipelines. An ML pipeline goes beyond the use of a collection of methods to encompass all stages of the data mining process, including data pre-processing (data cleaning, data transformations potentially including feature selection and extraction, but also other aspects of data curation such as data extraction and standardization, missing data imputation and data clinical validation [27]) and models’ post-processing, potentially including evaluation, implementation and the definition of interpretability and explainability processes [25]. Pipelines can also accommodate specific needs, such as those related to the analysis of “big data”, with their corresponding challenges of standardization and scalability. As described in [28], in a clinical oncology setting, this may require a research infrastructure for federated ML based on the *findable*, *accessible*, *interoperable*, and *reusable* (FAIR) principles. Alternatively, we can aspire to automate the ML pipeline definition using Automated ML (AutoML) principles, as in [29], where Su and co-workers used a Tree-based Pipeline Optimization Tool (TPOT) in the process of selecting radiomics features predictive of mutations associated with midline gliomas.

An example of an ML pipeline for the specific problem of differentiation of glioblastomas from single brain metastases based on MR spectroscopy (MRS) data can be found in [30]. In this same issue of Cancers, Pitarch and co-workers [31] describe an ML pipeline for glioma grading from MRI data with a focus on the trustworthiness of the predictions generated by the ML models. This entails robustly quantifying the uncertainty of the models regarding their predictions, as well as implementing procedures to avoid data leakage, thus avoiding the risk of unreliable conclusions. All of these can be seen as part of a quest to avoid the pitfalls of implementation of ML-based CDSS that result in the problems of limited reproducibility of analytical results in clinical practice that have been reported in recent studies [1].

As previously explained, the first stages of an ML pipeline, prior to the data modeling itself, involve data pre-processing, and this task may, in turn, involve many sub-problems. As an example of the potential diversity and complexity of this landscape, we comment here on a few recently selected contributions to the problem of feature engineering and extraction following just two particular and completely different approaches: statistical image feature engineering using radiomics and source extraction using ICA- and NMF-based methods.

Radiomics is an image transformation approach that aims to extract either hard-coded statistical or textural features based on expert domain knowledge or feature representations learned from data, often using DL methods. The former may include first-order statistics, size and shape-based features, image intensity histogram descriptors, image textural information, etc. The use of this method for the pre-processing of brain tumor images prior to the use of ML has been recently and exhaustively reviewed in [32]. From that review, it is clear that the predominant problem under analysis is diagnosis, with only a limited number of studies addressing prognosis, survival, and progression. The types of brain tumors under investigation are dominated by the most frequent classes. In particular, glioblastoma, either on its own or combined with metastasis as a *super-class* of aggressive tumors, is the subject of many studies, with some others also including other frequent super-classes such as low-grade glioma or meningioma, while minority tumor types and grades are only considered in a limited number of studies. Importantly, and related to our previous comments concerning scarce data availability, most of the studies reported in [32] work with very small sample sizes, often not reaching the barrier of 100 cases. The challenge posed by data scarcity is compounded by the fact that most of the studies extract Radiomic features in the hundreds if not the thousands. This means that the ratio of cases-to-features is extremely low, making the use of conventional ML classifiers very difficult. To alleviate this problem, most of the reviewed papers resort to different strategies for qualitative and quantitative feature selection. Image modalities under analysis are dominated by T1, T2, and FLAIR, with few exceptions (PET, or Diffusion- and Perfusion-Weighted Imaging). Most studies are shown to resort to the Area Under the ROC Curve (AUC) as a performance metric, which is a safe choice, as it is far more robust than plain accuracy for small and class-imbalanced datasets.

The use of radiomics as a data transformation strategy in pre-processing is facilitated by the existence of off-the-shelf software such as the open-source PyRadiomics package [33].

Source extraction methods have a very different analytical rationale for data dimensionality reduction as a pre-processing step. They do not achieve it through plain feature transformation, as in radiomics. Instead, they aim to find the underlying and unobserved sources of observed radiological data. In doing so, they achieve dimensionality reduction as a byproduct of a process that may provide insight into the generation of the images themselves.

The ICA technique [34] has a long history in medical applications, most notoriously for the analysis of electroencephalographic signals. Source extraction is natural in this context as a tool for spatially locating sources of the EEG from electric potentials measured in the skull surface. In ICA, we assume that the observed data can be expressed as a linear combination of sources that are estimated to be statistically independent or as independent as possible. This technique has mostly been applied to brain tumor segmentation, but some alternative recent studies have extended its possibilities to dynamic settings, such as that in [35], where dynamic contrast-enhanced MRI is analyzed using temporal ICA (tICA), and in [36], where probabilistic ICA is used for the analysis of dynamic susceptibility contrast (DSC) perfusion MRI.

The NMF technique [37], on the other hand, was originally devised for the extraction of sources from images and assumes data non-negativity but does not assume statistical independence. Data are still approximated by linear combinations of factors. Although NMF and variants of this family of methods have extensively been used for the pre-processing and analysis of MRS and MRS imaging (MRSI) signal [38,39], they have only scarcely been used for the pre-processing of MRI. Some outstanding exceptions include the work in [40] with hierarchical NMF for multi-parametric MRI and the recent proposal of a whole new architecture based on NMF called Factorizer [41], constructed by replacing the self-attention layer of a Vision Transformer (ViT, [42]) block with NMF-based modules.

The technical details of ICA and NMF and their manifold variants are beyond the scope of this review and can be found elsewhere in the literature.

## 4. Deep Learning in Neuro-Oncology Data Analysis: A Review

In this section, we review existing recent literature to gather evidence about the advantages, challenges, and potential future directions in the use of DL techniques for supervised problems in neuro-oncology. Furthermore, we aim to provide insights into the current state-of-the-art methodologies, address their limitations, and identify areas for further research. Ultimately, our objective is to facilitate the development of robust, responsible, and applicable DL solutions that can effectively contribute to the field of neuro-oncology.

### 4.1. Overview of the Main DL Methods of Interest

Recent advances in the DL field have brought about new possibilities in medical imaging analysis and diagnosis. One of its arguably most successful models is Convolutional Neural Networks (CNNs), a widely used type of DL algorithm, well known for its ability to capture spatial correlations within image pixel data hierarchically. They have shown promise in medical imaging tasks [43,44,45], enabling improved tumor detection, classification, and prognosis assessment. The input data of a CNN is represented as a tensor with dimensions in the format of *(channels, depth, height, width)*. Notably, the “depth” dimension is specific to 3D images and not applicable to 2D data, and “height” and “width” correspond to the image’s spatial dimensions. In practical terms, the number of channels for color images is translated into three, representing Red, Green, and Blue (RGB) components, while gray-scale images consist of a single channel. The most characteristic operation in a CNN is called *convolution*, which gives the name to the convolutional layers. These layers capture spatial correlations by applying a set of filters or kernels across all areas of the input image data and compute the weighted sum, resulting in the generation of a feature map as an output. This feature map contains essential characteristics extracted by the actual layer and serves as the input for subsequent layers of processing. Another useful layer used in CNNs is the pooling layer. The pooling operation consists of downsampling the feature maps obtained from the convolution operation. The idea is to reduce the dimensionality without losing significant information. There are mainly two kinds of pooling: max-pooling and average-pooling. The outputs of convolutional layers are often passed through activation functions to introduce non-linearity. The most popular activation functions are ReLU, which inactivates negative values in the output through the formula f(x)=max(0,x); Sigmoid, which maps output values between 0 and 1 using the equation $f\left(x\right)=\frac{1}{1+e^{-x}}$; and SoftMax, which is the extension of Sigmoid for multi-class problems.

CNNs often consist of multiple layers that work together to learn hierarchical high-level image features. These layers progressively extract more abstract and complex information from the input image data. In the final step, the last feature map is passed through a fully connected layer, resulting in a one-dimensional vector. To obtain the class probabilities, Sigmoid or SoftMax are applied to this vector.

Several networks have made significant contributions to the world of DL. AlexNet [46], GoogLeNet [47], InceptionNet, VGGNet [48], ResNet [49], DenseNet [50], and EfficientNet [51] are among the most widely used CNNs to extract patterns from medical imaging.

DL models are considered data-hungry since they require substantial amounts of data for effective training. In the realm of medical data analysis, a primary challenge, as previously mentioned, is the inherent data scarcity and class imbalance. Common solutions to address this challenge include the application of data-augmentation (DA) methods and transfer-learning (TL) techniques.

Data Augmentation techniques are a crucial strategy to mitigate the challenge of limited annotated data in medical image analysis. These methods encompass a range of transformations applied to existing images, effectively expanding the dataset in terms of both size and diversity. Former approaches involve a wide range of geometric modifications such as rotation, scaling, flipping, cropping, zooming, or color changes. Beyond traditional augmentations, advanced methods like Generative Adversarial Networks (GANs) [52] are used to generate new synthetic and realistic examples.

The idea behind TL is to leverage pre-trained models, typically trained in large and diverse datasets, and adapt them for the specific task at hand, for which we might not have such a representative sample. Widely used pre-trained CNNs, such as ImageNet [53] or MS-COCO [54], have been originally developed from 2D large-scale datasets. However, a notable challenge when dealing with medical image data is the limited availability of large and diverse 3D datasets for universal pre-training [55]. Transferring the knowledge acquired from the 2D to the 3D domain proves to be a non-trivial task, primarily due to the fundamental differences in data structure and representation between these two contexts. To tackle this challenge and address the limitation of limited data, a broadly used strategy is to decompose 3D volumes into individual 2D slices within a determined anatomical plane. However, the decomposition of 3D volumes into individual 2D slices introduces a potential data leakage concern. This issue arises when 2D slices from the same individual inadvertently end up in both the training and testing datasets in an analytical pipeline. Such data leakage can lead to overestimations of model performance and affect the validity of experimental results. In addition, it is important to note that this approach comes with the trade-off of losing the 3D context present in the original data.

Recent efforts have aimed at overcoming these challenges. Banerjee et al. [56] classified low-grade glioma (LGG)and high-grade glioma (HGG) multi-sequence brain MRIs from TCGA and BraTS2017 data using multiple slice-based approaches. In their work, they provided a comparison of the performance obtained with CNNs trained from scratch on 2D image patches (PatchNet), entire 2D slices (SliceNet), and multi-planar slices through a final ensemble method that averages the classification obtained from each anatomical view (VolumeNet). The classification obtained with these models is also compared with pre-trained VGGNet and ResNet on ImageNet. The multi-planar method outperformed the rest of the approaches with an accuracy of 94.74%, and the lowest accuracy (68.07%) was obtained with pre-trained VGGNet. Unfortunately, TCGA and BraTS data share some patient data, which could involve an overlap between training and testing samples and hence be prone to data leakage. Ding et al. [57] combined radiomics and DL features using 2D pre-trained CNNs using single-plane images and performing a subsequent multi-planar fusion. VGG16, in combination with radiomics and RF, achieved the highest accuracy of 80% when combining the information obtained from the three views. Even though the multi-planar approach processes the information gathered from the axial, coronal, and sagittal views, it is still essentially a 2.5D approach, weak at fully capturing 3D contexts. Zhuge et al. [58] presented a properly native 3D CNN for tumor segmentation and subsequent binary glioma grade classification and compared it with a pre-trained 2D ResNet50 on ImageNet with previous tumor detection, employing a Mask R-CNN. The results of the 3D approach were slightly higher than the 2D ones, reporting 97.10% and 96.30% accuracy, respectively. In their study, Chatterjee et al. [59] explored the role of (2+1)D, mixed 2D–3D, and native 3D convolutions based on ResNet. This study highlights the effectiveness of mixed 2D–3D convolutions, achieving an accuracy of 96.98%, surpassing both the (2+1)D and the pure 3D approaches. Furthermore, the use of pre-trained networks demonstrated enhanced performance in the spatial models, yet, intriguingly, the pure 3D model performed better when trained from scratch. A study conducted by Yang et al. [55] introduced ACS convolutions, a novel approach that facilitates TL from models pre-trained on large publicly accessible 2D datasets. In this method, 2D convolutions are divided by channel into three parts and applied separately to the three anatomical views (axial, coronal, and sagittal). The effectiveness of this approach was demonstrated using a publicly available nodule dataset. Subsequently, Baheti et al. [60] further advanced the application of ACS convolutions by showcasing their enhanced performance on 3D MRI brain tumor data. Their study provides evidence of notable improvements in both segmentation and radiogenomic classification tasks.

### 4.2. Publicly Available Datasets

Access to large and high-quality datasets plays a crucial role in the development and evaluation of robust DL classification algorithms. This section aims to provide a comprehensive review of several publicly accessible datasets that have been widely used in brain tumor classification tasks and DL research. These datasets encompass diverse tumor types, imaging modalities, and annotated labels, facilitating the advancement of computational methods for accurate tumor classification.

Table 1 provides a detailed overview of the most frequently used datasets in the literature.

The Brain Tumor Segmentation Challenge (BraTS) and The Computational Precision Medicine: Radiology-Pathology Challenge on Brain Tumor Classification (CPM-RadPath) datasets were created for two popular challenges held at the MICCAI (Medical Image Computing and Computer Assisted Intervention) Conference.

The BraTS Challenge [61] was initially developed in 2012 to benchmark tumor segmentation methods distinguishing glioblastoma from “lower grades”. Notably, this challenge provides not only MRI data but also clinical labels, including a binary classification of glioma grades. Even though their definition does not fully align with WHO’s terminology, they include grades 2 and 3 when referring to “lower grades”.

Throughout the years, the BraTS Challenge has continually evolved, expanding to include additional tasks and diverse datasets. In 2017, the dataset was enriched by integrating data from the TCIA repository, specifically including samples from the TCGA-LGG [71] and TCGA-GBM [70] datasets. It is worth noting that TCGA-LGG data provides labels to differentiate between gliomas of grades 2 and 3. Although the primary focus of the BraTS Challenge has traditionally centered on automated brain tumor segmentation, it has grown to become a widely adopted resource for brain tumor grade classification. Recent challenges have included tasks such as survival prediction and genetic classification, and the 2023 challenge even included image synthesis tasks.

CPM-RadPath [62] from 2019 was designed to evaluate brain tumor classification algorithms in three classes, taking into account the WHO classification of 2016: A (astrocytomas grades II and III, IDH-mutant), O (oligodendroglioma grades II and III, IDH-mutant, 1p/19q codeleted) and G (Glioblastoma and diffuse astrocytic glioma with molecular features of glioblastoma, IDH-wildtype (Grade IV)), interestingly grouping the anaplasic with the low grades in the A and O classes.

This challenge provides participants with paired radiology scans and digitized histopathology images. It is worth noting that the data provided by these challenges are distributed *after* pre-processing, involving co-registered to the same anatomical template, interpolated to a consistent resolution of 1 mm^3^, and skull-stripped.

The datasets under consideration encompass a variety of MRI modalities. Specifically, BraTS, CPM-RadPath, REMBRANDT, and TCGA comprise images from four key modalities: T1, T1 post-contrast weighted (T1c), T2-weighted, and Fluid Attenuated Inversion Recovery (FLAIR). The IXI dataset provides not only T1 and T2 but also Proton Density (PD) and Diffusion-weighted (DW) images. Notably, images on Figshare are limited to the T1c modality, while datasets from Kaggle and Radiopaedia lack this information.

The images in the BraTS, CPM-RadPath, IXI, REMBRANDT, and TCGA datasets are stored in 3D structures using widely used medical image formats, specifically NIfTI or DICOM. In contrast, datasets sourced from Kaggle consist of 2D images in PNG format. Notably, Figshare contains 2D images in MATLAB data format. In the Figshare data repository, images are provided alongside a 5-fold CV split at the patient level to mitigate the risk of data leakage. The use of this split ensures that no patient is inadvertently present in both training and testing sets, thus preventing leakage. Moreover, this dataset comprises multiple 2D slices from the same patient in the three distinct anatomical perspectives. Conversely, the datasets sourced from the Kaggle repository lack patient identifier information, making it challenging to ascertain if images are from unique patients or to trace the origin of the data.

Figure 1 summarizes the prevalence of dataset usage in the reviewed literature, including public and private datasets. Datasets that appear in two or fewer papers are grouped under the “Others” category.

It is worth highlighting that over 85% of the papers reviewed in this analysis make use of public datasets. It is essential to acknowledge that the sample sizes of the datasets, in general, are roughly in the hundreds range. This limited sample size can pose challenges in drawing robust and generalizable conclusions, which is a notable concern within the ML healthcare domain. Addressing the need for larger and more diverse datasets, as previously discussed, is an ongoing challenge in this field.

### 4.3. Literature Review

Various online repositories of scientific research articles, including PubMed, Google Scholar, and Scopus, were utilized to collect pertinent papers for this review section. The selection was restricted to the years 2018–2023. More specifically, only articles published prior to 30 June 2023 were taken into consideration. The document type was restricted to journal or conference papers. The focal keywords were centered on classifying brain tumors from pre-operative MRI images using DL techniques. While refining our choices, we excluded publications with ambiguous data explanations or lacking methodology details, as the utmost priority was placed on guaranteeing the strength and acuity of our conclusions. An initial identification process yielded a total of 555 papers, with 146 papers remaining after the screening procedure. Figure 2 depicts the distribution of these papers across the years under review, shedding some light on the temporal evolution of research in this domain.

In the subsequent analysis, we provide comprehensive insights into the data sources and methodologies employed in the examined papers. Table A1 offers a detailed overview of the datasets, focusing on essential aspects such as the dimensionality of the images, sample size, MRI details, and pre-processing methods used. Table A2 delves into the specifications of the employed DL models, highlighting the brain tumor classification task, data partitioning, architecture, and the reported performance metrics. These tables contribute to a comprehensive understanding of the methodologies employed in the reviewed literature. Table A1 and Table A2 exclusively display the information available from the original authors in the analyzed papers. Any omissions in the table reflect the absence of such details as provided by the original authors in the surveyed papers. Notice that several papers are marked with an asterisk (*), which denotes that not all models have been reported in our tables due to the extensive array of results reported by the authors. Especially in these cases, we recommend readers refer to the original papers for a comprehensive overview of findings.

In this review, we focus on the differentiation of primary brain tumor types, with particular attention to gliomas due to their aggressive nature. Among the 146 examined papers in this section, some address multiple tasks concurrently. Specifically, 77 focused on distinguishing primary brain tumor types, while 27 aimed to identify tumorous images from images of healthy patients. Furthermore, the pursuit of accurate glioma grade classification is assessed in 66 papers, with 41 of them focusing on the binary distinction between low (grades II and III) and high-grade (glioblastoma, grade IV) gliomas. Note that the question asked by these 41 works does not correspond to any of the canonical releases of the WHO classification of brain tumors, as III is, in fact, high-grade. In this sense, such a grouping would facilitate the achievement of good performance results by grouping entities that are more prone not to show contrast enhancement, in contrast to glioblastoma, which always will show contrast enhancement [72]. Additionally, 12 studies delve into the distinction of glioma subtypes.

As previously highlighted, pre-processing techniques are pivotal in medical image processing. Among the 146 papers analyzed in this review, a substantial 80% of them provide insights into the specific pre-processing methodologies that were employed. Within this subset, it was observed that 35% employed registration techniques involving registration to a common anatomical template and co-registration to the same MRI modality. Furthermore, 40% employed segmentation as a critical step to isolate the brain from the surrounding skull structures. Notably, nearly half of the papers embraced normalization techniques to standardize the intensity of the image data before it was fed to the models. Additionally, 30% of the papers undertook the task of brain tumor extraction through methods such as bounding box delineation or tumor segmentation. Moreover, 15% of the papers employed pre-processing integrated image enhancement techniques to improve the contrast and visibility of crucial anatomical structures. In several studies [73,74,75], researchers investigated the advantages of utilizing the tumor area as opposed to the entire image, highlighting the significant benefits of concentrating on the tumor region rather than the entire image.

In the realm of medical research, the size and diversity of the training data sample stand as fundamental factors that substantially influence the performance, generalizability, and robustness of ML models. Several studies have explored the impact of varying the size of the training data sample on model performance [76,77,78,79,80,81,82,83,84]. Their findings highlight the value of ensuring that a substantial volume of data is available for training, as it significantly contributes to the model’s ability to make more accurate and reliable predictions.

Regarding addressing the limitation of data scarcity, approximately 60% of the examined studies employed DA techniques, and 40% incorporated TL in a 2D domain as a viable solution. Several of these investigations [85,86,87,88,89,90,91,92] have demonstrated the advantages of increasing both the quantity and variability of the samples through the inclusion of augmented images. Applying traditional DA techniques, such as geometric variations from original images, was the most widely used strategy, while only a few studies opted for the use of DL generative models [89,93,94,95]. Several studies [73,92,96,97,98,99] have integrated DA as an oversampling technique to address the problem of imbalanced data in the context of brain tumor classification. Furthermore, other works have explored the inclusion of multi-view 2D slices from axial, coronal, and sagittal planes, in addition to employing image flipping and rotations to augment the dataset [100]. Pre-trained models have demonstrated performance enhancements in the classification of glioma grades in several studies [79,101,102]. However, it is noteworthy that not all investigations have reported equivalent advantages when employing pre-trained models to discriminate between healthy and tumorous samples [103] or to differentiate tumor types [104]. These variations in findings underscore the complexity of the observed performance disparities, which may not be solely ascribed to the classification task itself but may also be influenced by intrinsic dataset variations.

The ability of CNNs to automatically extract meaningful features from brain MRI images, as opposed to the conventional need for manual feature engineering in certain ML algorithms like RF, GrB, and SVM, has been emphasized by numerous studies. These studies underscore the potential of CNNs in revolutionizing the landscape of MRI feature extraction for enhanced accuracy and efficiency in brain tumor classification [105,106,107,108,109,110,111]. Most of the reviewed papers (approximately 60%) utilized established state-of-the-art CNN architectures to obtain brain tumor classification. Among these, ResNet and VGGNet backbones were the most prevalent choices, closely followed by AlexNet, GoogLeNet, and Inception. In contrast, the remaining 40% of the papers concentrated on enhancing brain tumor classification by introducing novel model architectures. The inherent black-box nature of CNNs highlights the importance of delving into the comprehension of their predictions, especially in a medical context. Several studies within our review [74,112,113,114,115,116,117] have applied post-processing explainability tools to validate that the network’s decision-making process aligns with the intended diagnostic criteria, therefore enhancing the reliability of CNN-based medical applications.

Additionally, selected studies [57,118,119] explored the synergies of ensemble learning by combining the outputs of radiomics and DL models. Another interesting area of research has considered the opportunity of incorporating ML classifiers as the final layer in CNNs, effectively bypassing the traditional SoftMax layer [76,96,99,103,119,120,121,122,123,124,125,126,127,128].

The integration of information from various data sources has garnered growing interest in the medical field. Brain tumors, due to their distinct features both at the histopathological and radiological level, have motivated numerous studies to explore the synergy between whole slide imaging (WSI) and MRI data [97,129,130,131,132]. These investigations consistently highlight the richer information content in WSI as compared to MRI. However, they also reveal that combining data from both sources leads to improved overall performance in brain tumor characterization. Ensemble learning methods have shown promise in not only integrating information from diverse data types but also in combining predictions from multiple DL models on MRI to improve overall performance [75,82,91,107,108,116,122,133,134,135,136,137,138,139]. As brain tumor diagnosis and prognosis are significantly linked to genetic factors, several studies have undertaken efforts to explore the capabilities of DL models in extracting meaningful MRI features for the classification of these genetic frameworks [56,83,93,100,117,140,141].

Although brain MRIs inherently capture 3D data, a notable observation is that over 80% of the studies conducted their analyses within a 2D domain, focusing on 2D MRI slices. Nonetheless, some investigations have actively explored the significance of incorporating 3D volumetric information into the realm of brain tumor classification [56,58,59,74,97,98,100,112,117,129,130,131,132,135,140,141,142,143,144,145,146,147,148]. Although 3D volumes inherently capture information from the three anatomical planes, 2D slices are restricted to a specific view. Notably, among the studies that adopted a 2D approach, only 44% provided details about the chosen anatomical plane. Among this subset, more than 50% utilized axial, coronal, and sagittal views, while over 40% exclusively employed axial views.

Similarly, close to 70% of the reviewed studies disclosed the MRI modalities utilized for the analysis. Among these, close to 50% exclusively employed the T1c sequence, while 26% used a combination of T1c, T1, T2, and FLAIR sequences, 12% used three sequences, and the rest chose one sequence. Various strategies were employed to integrate information from multiple modalities. The prevalent method involved fusing them as input channels, comparable to the treatment of channels in RGB images. In their study, Ge et al. [100] evaluated the sensitivity of T1c, T2, and FLAIR modalities in glioma grade classification. Their investigation highlighted the T1c sequence as the most informative among these modalities. To further enhance the classification performance, they incorporated information from each source using an aggregation layer within the network architecture. Subsequently, similar ensemble learning approaches were adopted by Gutta et al. [106], Hussain et al. [148], Rui et al. [149]. Notably, Guo et al. [150] directly compared the performance of a modality-fusion approach, where the four MRI modalities were concatenated as a four-channel input, with a decision-fusion approach, where final predictions were derived through a linear weighted sum from the probabilities obtained through four independent pre-trained unimodal models. This study reinforced the notion of the T1c modality’s significance in glioma subtype classification. Moreover, it revealed that any multimodal approach consistently outperformed unimodal models, with the decision-ensemble approach emerging as the most effective strategy.

As previously discussed, decomposing 3D volumes into individual 2D slices may introduce the potential for data leakage. Maintaining the reliability of the analysis is crucial for obtaining robust and trustworthy findings. It is worth noting, however, that only a limited number of studies that use multiple 2D slices [56,76,77,79,93,100,101,106,107,115,118,126,135,149,151,152,153,154,155,156,157], explicitly detailed their approach to data splitting at the patient level, addressing this critical concern. Remarkably, an insightful comparison was carried out in the work of Badža and Barjaktarović [158] between data-splitting strategies at the patient and image levels. The findings elucidate that an image-wise approach yields accuracy results as high as 96% for brain tumor type classification, while a patient-level split demonstrates a higher degree of reliability with an accuracy of 88%. These results underscore the critical importance of utilizing a patient-wise training approach to assess the model’s generalization capacity. Similarly, Ghassemi et al. [85], Ismael et al. [159] also provided evidence of superior performance when using an image-wise split, further reinforcing the importance of thoughtful data splitting. It is also important to note that 3D models operate on complete 3D volumes and are inherently structured at the patient level. This approach substantially reduces the likelihood of data leakage, therefore enhancing the reliability of the analysis and ensuring that the results faithfully represent the model’s performance. This aspect may provide a valuable perspective when interpreting differences in accuracy between 3D and 2D models.

The predominant approach for data partitioning in the examined papers involves the use of hold-out validation with training and validation sets. This was followed by the adoption of K-fold cross-validation, which enhances the robustness of model evaluation. A less frequently employed method was the three-way split, which includes training, validation, and testing sets. In total, only 36% of the studies assessed their final results using an independent test set. Decuyper et al. [140], Gilanie et al. [152], Alanazi et al. [160] took a step further by assessing the generalizability and robustness of their models using external validation sets. Although the authors of the Figshare dataset thoughtfully included a 5-fold CV setup alongside the data to promote comparability and reproducibility, it is still important to remark here that a substantial majority of studies continue to prefer custom data partitioning methods.

## 5. Machine Learning Applications to Ultra-Low Field Imaging

A completely different area of application of ML to neuroradiology has recently emerged with the availability of ultra-low field magnetic resonance imaging devices for point-of-care applications, typically with <0.1 T permanent magnets [161,162,163]. In the 0.055 T implementation described by Liu et al. [11], DL was used to improve the quality of the acquisition by detecting and canceling external electromagnetic interference (EMI) signals, eliminating the need for radio-frequency shielded rooms. They compared the results of the DL EMI cancelation in 13 patients with brain tumors, both in the 0.055 T and in another 3 T machine, on same-day acquisitions, finding that it was possible to identify the different tumor types. Please note that these processes are, in fact, a completely different use of DL for data pre-processing to those reviewed in Section 3.

Another example is the *Hyperfine* system, which received FDA clearance in 2020 for brain imaging and in 2021 (K212456) for DL-image reconstruction to enhance the quality of the generated images. In particular, DL is used as part of the image reconstruction pipeline of T1, T2, and FLAIR images. There are two DL steps: the first one is a so-called DL gridding, where the undersampled k-space data are transformed into images not by Fourier transformations but with DL. The transformed images are then combined, and a final post-processing DL step is applied to eliminate noise. However, no details about the specific algorithms are provided. Although the main application seems to be in the neurocritical setting [164], this system is beginning to be compared with the imaging quality at higher fields at different stages, with a particular interest in the early post-operative monitoring after surgical resection (e.g., [165,166]). It is to be expected that *Hyperfine* brain tumor applications will emerge soon, for example, through the partnership with The Brain Tumor Foundation, to provide the general population with free brain scans.

## 6. Conclusions

Neuro-oncological radiology relies on non-invasive data acquisition, which makes it the ideal target of data-centered analytics and places it at the forefront of ML-applied developments. In this review paper, we have focused on the most successful instantiation of ML currently, namely DL, and its use for the analysis of imaging data. Emphasis has been put on the fact that DL methods must be seen as only part of analytical pipelines, in which data pre-processing plays a key role.

Promoting the responsible utilization of clinical data is of utmost importance when striving to establish trustworthy conclusions. A fundamental step in this endeavor is the comprehensive disclosure of both the data used and the analytical procedures undertaken. Such transparency not only fosters greater trust in research outcomes but also amplifies the generalizability and reproducibility of the findings. This, in turn, plays a pivotal role in advancing AI-driven solutions in the clinical pathway. Most DL-based analytical solutions depend, to a great extent, not only on data quality but also quantity. For this reason, we argue that the main challenge facing the use of DL in the radiological imaging setting is precisely the creation of sizeable curated image databases for the different problems at hand.

## Figures and Tables

**Figure 1 cancers-16-00300-f001:**
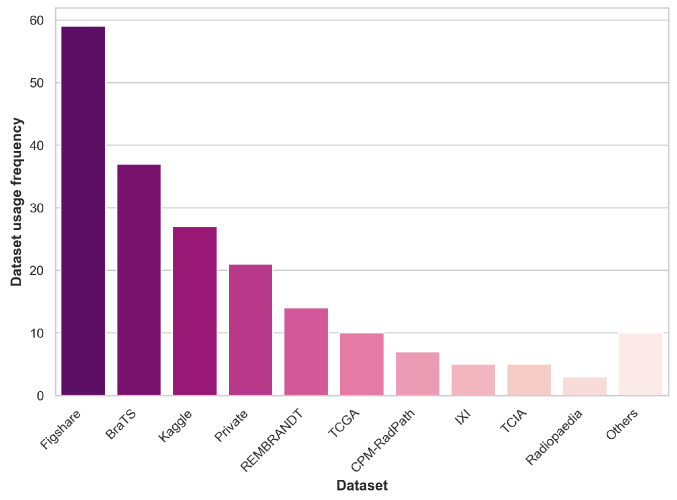
Dataset usage prevalence across the reviewed literature.

**Figure 2 cancers-16-00300-f002:**
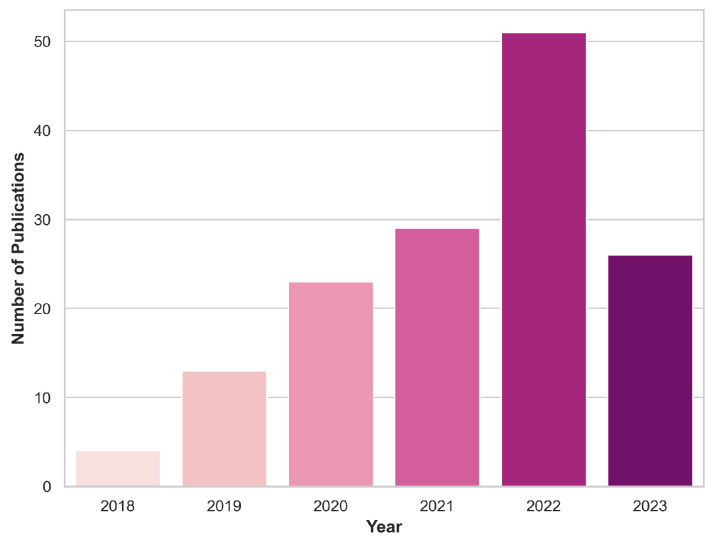
Yearly inclusion of articles in this review that focus on classifying brain tumors using DL and MRI scans.

**Table 1 cancers-16-00300-t001:** An overview of publicly available MRI datasets for brain tumor classification benchmarking.

Dataset		Categories	Dim.	Sample Size	MRI Modalities
BraTS [61]	2020	Low-Grade Glioma (LGG) High-Grade Glioma (HGG)	3D	369 (LGG: 76, HGG: 293)	T1, T1c, T2, FLAIR
	2019		3D	335 (LGG: 76, HGG: 259)	T1, T1c, T2, FLAIR
	2018		3D	284 (LGG: 75, HGG: 209)	T1, T1c, T2, FLAIR
	2017		3D	285 (LGG: 75, HGG: 210)	T1, T1c, T2, FLAIR
	2015		3D	274 (LGG: 54, HGG: 220)	T1, T1c, T2, FLAIR
	2013		3D	30 (LGG: 10, HGG: 20)	T1, T1c, T2, FLAIR
	2012		3D	30 (LGG: 10, HGG: 20)	T1, T1c, T2, FLAIR
CPM-RadPath [62]		Astrocytoma (AS) IDH-mutant Oligodendroglioma (OG) IDH-mutant 1p/19q codeletion Glioblastoma (GB) IDH-wildtype	3D	Training: 221 (AS: 54, OG: 34, GB: 133) [unseen sets] Val: 35, Test: 73	T1, T1c, T2, FLAIR
Figshare [63]		Meningioma (MN), Glioma (GL), Pituitary (PT)	2D	233 (MN: 82, GL: 89, PT: 62)	T1c
IXI [64]		Healthy	3D	600	T1, T2, PD, DW
Kaggle-I [65]		Healthy (H), Tumor (T)	2D	3000 (H: 1500, T: 1500)	-
Kaggle-II [66]		Healthy (H), Meningioma (MN), Glioma (GL), Pituitary (PT)	2D	3264 (H: 500, MN: 937, GL: 926, PT: 901)	-
Kaggle-III [67]		Healthy (H), Tumor (T)	2D	253 (H: 98, T: 155)	-
Radiopaedia [68]		-	-	-	-
REMBRANDT [69]		Oligodendroglioma (OG), Astroctyoma (AS), Glioblastoma (GB)	3D	111 (OG: 21, AS: 47, GB: 44)	T1, T1c, T2, FLAIR
		Grade II (G.II), Grade III (G.III), Grade IV (G.IV)		109 (G.II: 44, G.III:24, G.IV: 44)	
TCGA-GBM [70]		Glioblastoma	3D	262	T1, T1c, T2, FLAIR
TCGA-LGG [71]		Grade II (G.II), Grade III (G.III)	3D	197 (G.II: 100, G.III: 96, discrepancy: 1)	T1, T1c, T2, FLAIR
		Astroctyoma (AS), Oligodendroglioma (OG), Oligoastrocytoma (OAS)		197 (AS: 64, OG: 86, OAS: 47)	

DW: Diffusion-weighted, FLAIR: Fluid Attenuated Inversion Recovery, PD: Proton Density, T1c: contrast-enhanced T1 weighted.

## Data Availability

Data sharing is not applicable.

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
