# Peer review of "Advances in the Use of Deep Learning for the Analysis of Magnetic Resonance Image in Neuro-Oncology"

_cancers, 2024, doi:10.3390/cancers16020300_

Round 1

Reviewer 1 Report

Comments and Suggestions for Authors

The Authors have done a good job.

I do not have any comments or suggestions. 

Author Response

The authors thank reviewer 1 for her/his comments on the previous version of the manuscript.

Reviewer 2 Report

Comments and Suggestions for Authors

The paper deals on machine learning and artificial intelligence applied to the neuro-oncologic filed. The argument is very actual and in the interest of the general audience. The paper is very well written and offers a wide and comprehensive review of the field. The review of the letterature is complete and tables are representative. Concepts are well designed and organized in the structure of the review. This article can represent a good starting point for approaching this important topic for a large audience.  I have not substantial suggestions or critics to this work that can be considered for the publication in cancers.

Comments on the Quality of English Language

Quality of English is quite good and the paper is easily readable and understandable

Author Response

The authors thank reviewer 2 for her/his comments on the previous version of the manuscript.

Reviewer 3 Report

Comments and Suggestions for Authors

The authors performed a review on Deep Learning applications in neuro-oncology.

Overall, the paper is well written, however, I have some minor suggestions (in yellow) in the attached file.

1. In the abstract, it would be better if 

specific instantiations of the use of Deep Learning in neuro-oncology

2. When mentioning RANO, you should explain and include the following references:

If the lesion developed within the first 3–6 months after radiation therapy, if it is in the radiation field (inside the 80% isodose line), and especially if it presents as a pattern of enhancement related to radiation-induced necrosis, ie, “Swiss cheese” or “soap bubble” enhancement

Kumar AJ. Malignant gliomas: MR imaging spectrum of radiation therapy- and chemotherapy-induced necrosis of the brain after treatment. Radiology 2000;217:377–84

Also, with 100 pseudoresponse in those patients treated with antiangiogenics in countries where these are 101 approved [16,17]. Antiangiogenic agents, like bevacizumab, are designed to block the VEGF effect. The mechanism of action may be related to decreased blood supply to the tumor and normalization of tumor vessels, which display increased permeability.These agents are associated with high radiologic responses if we evaluate only the contrast enhancement.

Reardon DA. Glioblastoma multiforme: an emerging paradigm of anti-VEGF therapy. Expert Opin Biol Ther 2008;8:541–53.

3. Include:

Three different methods of ML: support vector machine (SVM), k-nearest neighbors (kNN), and Random Forest can be employed. Those three methods belong to a supervised class of ML. All supervised learning methods utilized texture features extracted from regions of interest. All three methods used the textural features to train and then test data in a 10-fold cross-validation procedure. The training set was composed of 75% of all the input data and the test with 25% of input data.

Wernick MN. Machine learning in medical imaging. IEEE Signal Process Mag. 2010;27(4):25-38.

Alves AFF Inflammatory lesions and brain tumors: is it possible to differentiate them based on texture features in magnetic resonance imaging? J Venom Anim Toxins Incl Trop Dis. 2020 Sep 4;26:e20200011. 

4. and FLAIR, with exceptions (PET, or Diffusion

5. IN fact, grade 3 is a high grade glioma.

6. Grade I was not included? Why not?. mention why.

7. sensitivity of T1c, T2, and FLAIR

8. It is important to consider that an unsupervised approach for abnormal asymmetry detection based on supervoxel segmentation with the SymmISF method to extract symmetrical supervoxels in the brain can be employed. The proposed approach, named SAAD, was validated on 3T MR-T1 images of controls and patients with lesions, achieving much better detection rate and a drastically lower false positive rate compared to a DL autoencoder approach.

Martins SB. A Supervoxel-Based Approach for Unsupervised Abnormal Asymmetry Detection in MR Images of the Brain.  2019 IEEE 16th International Symposium on Biomedical Imaging (ISBI 2019), Venice, Italy, 2019, pp. 882-885,

Comments on the Quality of English Language

English is good.
